# Bee Venom Induces Acute Inflammation through a H_2_O_2_-Mediated System That Utilizes Superoxide Dismutase

**DOI:** 10.3390/toxins14080558

**Published:** 2022-08-17

**Authors:** Kwang-Sik Lee, Bo-Yeon Kim, Min-Ji Park, Yijie Deng, Jin-Myung Kim, Yun-Hui Kim, Eun-Jee Heo, Hyung-Joo Yoon, Kyeong-Yong Lee, Yong-Soo Choi, Byung-Rae Jin

**Affiliations:** 1Department of Applied Biology, College of Natural Resources and Life Science, Dong-A University, Busan 49315, Korea; 2Department of Agricultural Biology, National Academy of Agricultural Science, Wanju 55365, Korea

**Keywords:** bee venom, arthropod venom, envenoming, superoxide dismutase, hydrogen peroxide, acute inflammation

## Abstract

Venoms from venomous arthropods, including bees, typically induce an immediate local inflammatory response; however, how venoms acutely elicit inflammatory response and which components induce an inflammatory response remain unknown. Moreover, the presence of superoxide dismutase (SOD3) in venom and its functional link to the acute inflammatory response has not been determined to date. Here, we confirmed that SOD3 in bee venom (bvSOD3) acts as an inducer of H_2_O_2_ production to promote acute inflammatory responses. In mouse models, exogenous bvSOD3 rapidly induced H_2_O_2_ overproduction through superoxides that are endogenously produced by melittin and phospholipase A_2_, which then upregulated caspase-1 activation and proinflammatory molecule secretion and promoted an acute inflammatory response. We also showed that the relatively severe noxious effect of bvSOD3 elevated a type 2 immune response and bvSOD3 immunization protected against venom-induced inflammation. Our findings provide a novel view of the mechanism underlying bee venom-induced acute inflammation and offer a new approach to therapeutic treatments for bee envenoming and bee venom preparations for venom therapy/immunotherapy.

## 1. Introduction

Incidences of human envenoming by bee stings are a common health problem worldwide. Bee venom typically causes nonallergic reactions that elicit an immediate local inflammatory response. However, in certain subjects, bee venom can also cause severe allergic reactions due to an immediate hypersensitivity-induced reaction that leads to anaphylaxis [1]. As a therapy, bee venom is subcutaneously administered to treat various disorders, including arthritic rheumatism, pain, cancerous tumors, and skin diseases [2,3,4,5]. Although nociceptive and inflammatory responses are common limitations of bee venom therapy (VT) [6], the administration of bee venom through alternative therapeutic methods exerts antinociceptive and anti-inflammatory effects [2,4]. Moreover, bee venom immunotherapy (VIT) is the most effective treatment for reducing the risk of allergic reactions [7,8]. Therefore, understanding how bee venom induces acute inflammatory responses is essential to improving the treatment outcomes of VT and VIT, as well as the therapeutic treatments for envenoming [6,8,9].

Bee venom is a complex mixture of toxic components, including a variety of enzymes, peptides, biogenic amines, and nonpeptide components, which display various biological, pharmacological, and toxicological activities [2,6,10,11,12]. The major toxic compound of bee venom is melittin, which possesses lytic activity, followed by the predominant allergen phospholipase A_2_ (PLA_2_), which has inflammatory and nociceptive effects [13,14,15]. Previous studies have demonstrated that bee venom and melittin activate the release of oxygen radicals and the transcription of proinflammatory genes [16] and induce caspase-1 activation and interleukin (IL)-1β secretion [17] and that PLA_2_ also triggers IL-1β release [17]. A general mechanism has been proposed, through which melittin and PLA_2_ induce inflammatory responses. Despite extensive investigation, previous studies have failed to identify the component that induces reactive oxygen species (ROS) in bee-venom-induced inflammation. Due to the lack of a firm understanding of the venom components, the mechanism underlying bee venom-induced acute inflammation remains incompletely understood. Specifically, the presence of superoxide dismutase (SOD) in the venom of venomous arthropods, including bees, has not been previously demonstrated. As the first line of defense against ROS, SOD is a key superoxide-scavenging antioxidant enzyme that catalyzes the dismutation of superoxide to H_2_O_2_ [18,19,20,21]. In contrast, H_2_O_2_ overproduction is involved in inflammation and in many regulatory cellular events [22,23,24,25,26]. H_2_O_2_ drives the onset of inflammation through NFκB activation [27]. Although the beneficial antioxidant role of SOD as a defense system against ROS is well-known, the detrimental effects of SOD remain largely unknown.

It is well-known that H_2_O_2_ acts as a mediator of immediate inflammation [22,23,24,25,26] and that bee venom typically causes immediate inflammation [1]. A recent study has proposed a mechanistic model for toxin synergism, which is related to cytotoxicity by cytotoxin/melittin and PLA_2_ complex formation [28]. However, neither the component that induces the bee-venom-mediated production of H_2_O_2_ nor the component that promotes bee-venom-induced immediate inflammation is known. Therefore, it could be surprising to provide the rationale and presence of novel components in bee venom and uncover the potential mechanisms underlying the induction of acute inflammation.

Here, we provide the first demonstration that venoms contain Cu,Zn-SOD (SOD3), an extracellular enzyme. As bee venom serves as a defensive weapon against human and vertebrate predators, we hypothesized that bee venom SOD3 (bvSOD3) serves as an ROS-based harm-inducing system in bee venom. We investigated the role of bvSOD3 in the bee venom-induced inflammatory responses in mouse models because SOD3 produces H_2_O_2_, which acts as a mediator of immediate inflammation [22,23,24,25,26]. Notably, we demonstrated that bvSOD3 functions as an inducer of H_2_O_2_ production to promote an acute inflammatory response. Finally, we demonstrate that the bvSOD3-mediated overproduction of H_2_O_2_ led to the induction of caspase-1 activation and proinflammatory molecule secretion, providing a novel view of the mechanism underlying the bee-venom-induced acute inflammatory response. Additionally, we implicate a type 2 immune response using the noxious effect of bvSOD3 and a protective immune response by bvSOD3 immunization.

## 2. Results

### 2.1. Bee and Arthropod Venoms Contain a Functional SOD3 Enzyme

To determine how venom elicits acute inflammatory responses, we focused on SOD3, a novel component of bee venom. Analyses of the enzymatic activities of bee venoms revealed that both honeybee (*Apis mellifera* and *A. cerana*) and bumblebee (*Bombus terrestris* and *B. ignitus*) venoms present the SOD3 enzyme, but not catalase or peroxidase enzymes (Figure 1A). Honeybee venom had relatively low SOD3 levels compared with bumblebee venom (Figure 1B). Using an anti-*A. mellifera* SOD3 (AmSOD3) antibody raised against a recombinant AmSOD3 protein produced in baculovirus-infected insect cells (Appendix A), we confirmed the presence of SOD3 in the venom gland and secreted venom (bvSOD3) (Figure 1C), which indicated that bvSOD3 is an extracellular SOD. Consistent with the identity in bvSOD3 protein sequences (Appendix A), the anti-AmSOD3 antibodies cross-reacted with the SOD3 enzymes in the venoms of *A. cerana*, *B. terrestris*, and *B. ignitus* (Figure 1D). As the inflammatory effects and components of bee venom have been extensively studied in recent decades [6,11,12,29], the evidence shows that bee venom contains a SOD3 enzyme as a novel venom component. This is a surprising new result. We further confirmed the presence of SOD activity in the venom of spiders (*Latrodectus mactans*), hornets (*Vespa mandarinia*), and centipedes (*Scolopendra subspinipes mutilans*) (Figure 1E). In addition to the previous results, showing the presence of secreted SOD3 in wasp venom [30], our data suggest that SOD3 is a conserved component of venoms from venomous arthropods.

### 2.2. bvSOD3 Induces Caspase-1 Activation and Proinflammatory Molecule Secretion via H_2_O_2_ Overproduction

As venom evolved as a defensive weapon against predators, we hypothesized that bvSOD3 acts as an ROS-based harm-inducing system in bee venom rather than as a defensive system against ROS. To address this issue, we prepared *A. mellifera* bee venom (AmV) containing blocked SOD3 (AmVΔSOD3) by immunoprecipitation with an anti-AmSOD3 antibody (Appendix A). The analysis of in vitro H_2_O_2_ production induced by bvSOD3 revealed that native AmV produced H_2_O_2_, whereas AmVΔSOD3 produced almost no H_2_O_2_ (Appendix A). These results indicated that the production of H_2_O_2_ induced by bee venom was dependent on SOD3, and thus define a specific role for SOD3 in bee venom. To determine whether bvSOD3 affects pathological outcomes, we injected mice with AmV and AmVΔSOD3. Surprisingly, compared with the mice injected with native AmV, the mice injected with AmVΔSOD3 showed a decreased immediate inflammatory response, as indicated by the reduced redness, swelling, and inflammatory mediator release in the injected inguinal tissues of the rear leg and the native AmV-injected mice exhibited a more severe inflammatory response (Figure 2A) and more severe damage in muscle tissues due to the increased levels of caspase-1 and apoptosis [17,31] (Figure 2B and Appendix A). Thus, bvSOD3 activity appears to promote an immediate local inflammatory response.

We subsequently assessed whether bvSOD3 induces H_2_O_2_ production in mice. The mice injected with AmVΔSOD3 showed a significantly decreased generation of ROS and H_2_O_2_ compared with mice injected with native AmV (Figure 2C,D). Importantly, these results indicated that H_2_O_2_ production was acutely increased in mice injected with bee venom and that the bee-venom-mediated overproduction of H_2_O_2_ was dependent on SOD3. In addition, the level of SOD activity in the AmV-injected mice was higher than that in the mice injected with AmVΔSOD3 (Appendix A), which suggested that bvSOD3 acts as an exogenous SOD. The overproduction of H_2_O_2_ mediated by exogenous bvSOD3 activity altered the noxious effects of venom injection, including local inflammation, although catalase and peroxidase activity in the mice increased in an H_2_O_2_-dependent manner (Appendix A) to allow for neutralization of tH_2_O_2_ toxicity. As excessive H_2_O_2_ production overwhelms cellular scavenging systems and causes acute inflammation and tissue damage [25,32], our data showed that bee venom, which contains SOD3, can promote inflammatory responses by producing significant amounts of H_2_O_2_, and these amounts were higher than those produced by the non-SOD3 components of bee venom. These results indicated that bvSOD3 does not play a role in the defense system against ROS. Instead, bvSOD3 might function as a detrimental system that utilizes ROS, which indicates that the immediate inflammatory response to venom is likely due to the deleterious effects of the SOD enzyme.

We further confirmed that bvSOD3-mediated H_2_O_2_ overproduction promoted the expression of proinflammatory mediators and cytokines, such as tumor necrosis factor (TNF)-α, cyclooxygenase (COX)-2, interleukin (IL)-1β, and IL-6, in mice (Figure 2E), which demonstrated bvSOD3′s critical role in promoting the immediate inflammatory response. Previous studies have shown that venoms from venomous animals [33,34,35], including bees [16,17], induce an inflammatory response in a proinflammatory mediator- and cytokine-dependent manner. We investigated the induction of proinflammatory molecules, which was attributed to bvSOD3. These results further demonstrated that bvSOD3 plays a pathological rather than beneficial role in bee-venom-induced inflammation. Furthermore, our findings, together with the observations that bvSOD3 promotes an immediate local inflammatory response and that other arthropod venoms exhibit SOD activity, suggest that arthropod-venom-induced acute inflammation is regulated in an SOD-dependent manner.

### 2.3. bvSOD3 Promotes Acute Induction of Inflammatory Responses

As we observed that the mice injected with recombinant AmSOD3 alone did not develop an inflammatory response or H_2_O_2_ production compared with control mice injected with PBS (Appendix A), we further examined the requirement for superoxide in the bvSOD3-mediated induction of H_2_O_2_ production. A combination of AmSOD3 and major venom components, such as melittin and PLA_2_, was required to induce the production of ROS, H_2_O_2_, and proinflammatory molecules in mice (Figure 3A–C). These treatments exerted more severe inflammatory effects than melittin and/or PLA_2_ alone (Figure 3D). Thus, bvSOD3, as an exogenous SOD, rapidly induced H_2_O_2_ overproduction by using superoxide endogenously produced by melittin and PLA_2_, and the resulting H_2_O_2_ upregulated caspase-1 activation and the secretion of proinflammatory molecules and led to an immediate inflammatory response (Figure 3E). Collectively, these results indicate that bvSOD3 promotes the acute induction of inflammatory responses driven by the toxin synergism of melittin and PLA_2_ [13,15,28], and thus provides a rationale for the presence of SOD3 in bee venom.

### 2.4. Noxious Effects of bvSOD3 Elevate Type 2 Immune Responses

In addition to the known mechanism of type 1 cytokine activation involved in envenomation, type 2 immune responses are induced as a host defense against venoms [17,36]. We observed the production of type 2 cytokines (IL-4 and IL-13) in mice injected with melittin and PLA_2_ (Figure 4A). Notably, bvSOD3 induced the production of type 2 cytokines in a similar manner to that of type 1 cytokines (Figure 4A,B). As tissue damage, H_2_O_2_, and the noxious effects of venoms lead to the induction of a type 2 immune response [17,36], our results showed that the elevation of type 2 immune responses was due to bvSOD3, which increased the noxious effects involved in the inflammatory response to envenomation. Thus, these data suggest that the elevation of type 2 innate immune responses to repair damaged muscles is a response to the increased noxious effects induced through the bvSOD3-mediated inflammatory response [17,37].

### 2.5. bvSOD3 Immunization Protects against Bee Venom-Induced Inflammation

As PLA_2_ immunization protects against PLA_2_-mediated toxicity [37], we assessed whether immunization with bvSOD3 protects against the noxious effects involved in the inflammatory response to envenomation through the induction of an antibody response. After challenge with AmV, the pathological effects of the production of H_2_O_2_ and proinflammatory molecules were determined in mice immunized through injection with AmV, AmVΔSOD3, or AmSOD3. The mice immunized with AmV showed a decreased inflammatory response compared with mice immunized with AmVΔSOD3 (Figure 5A), which suggested that the more effective protection against immediate inflammatory responses observed in the mice immunized with AmV was likely due to bvSOD3. Notably, the mice immunized with AmSOD3 were protected against the noxious effects of envenomation compared with unimmunized mice (Figure 5A). These data are consistent with the production of antibodies against AmSOD3 in the mice immunized with AmV or AmSOD3 (Figure 5B), which led to the decreased production of H_2_O_2_ and caspase-1 (Figure 5C,D) and subsequently reduced the levels of proinflammatory molecules and apoptosis [17,31] (Figure 5E and Appendix A). Thus, immunization with bvSOD3 induced protection against the bee-venom-induced inflammatory response, which indicated that bvSOD3 can act as an immunogen in a protective immune response to limit acute inflammation.

## 3. Discussion

Human injury caused by bees is one of the most common envenoming events [38]. Interestingly, the paradoxical aspects of the biological action of bee venom have been intensively studied, including nociceptive and inflammatory responses and antinociceptive and anti-inflammatory effects [6]. The common toxic effects of bee venom are known to induce nociceptive and immediate inflammatory responses. However, the mechanism underlying the induction of inflammation by bee venom is poorly understood. Herein, we provide possible evidence for an acute inflammatory response induced by bvSOD3 by first demonstrating that bee venom contains the SOD3 enzyme.

We found that venom from honeybees and bumblebees contained the SOD3 enzyme. Similar to wasp venom SOD3, which is secreted in venom [30], we confirmed that the SOD in bee venom was SOD3 by showing that little or no SOD activity was present in bee venom blocked by an SOD3-specific antibody. Thus, the SOD activity of bee venom was due to the presence of SOD3, demonstrating that bee venom contains a functional SOD3 enzyme alone, without catalase and peroxidase enzymes. Moreover, our findings that the venom of spiders, hornets, and centipedes exhibits SOD activity, suggesting the possibility of the presence of SOD3 as a conserved component of venoms from venomous arthropods.

Bee venom, which serves as a defensive weapon against human and vertebrate predators, induces an immediate local inflammatory response that causes acute redness and swelling at the site of the sting [6]. In the present study, we demonstrated that bee venom contains SOD3 and described a possible function of bvSOD3. We found that bvSOD3 induces H_2_O_2_ production in vivo. The levels of H_2_O_2_ and ROS (including H_2_O_2_) in tissue at the site of injection were acutely and significantly increased. Thus, bvSOD3 likely converts the oxygen radicals generated by bee venom components, such as melittin, to H_2_O_2_ [16]. Consistent with our hypothesis that bvSOD3 is involved in H_2_O_2_ overproduction, which leads to an immediate local inflammatory response [22,23,24,25,26,32], mice injected with AmVΔSOD3 did not show a significant immediate local inflammatory response, as indicated by redness, swelling, and nociception. Along with our other data, this result suggests that bvSOD3 acts as a H_2_O_2_ inducer, which drives an immediate local inflammatory response at the site of the sting. This observation supports the concept that the high levels of H_2_O_2_ produced by the activity of exogenous bvSOD3 may be injurious to tissues and lead to local inflammation [22,23,24,25]. Several deleterious consequences of H_2_O_2_ overproduction have previously been attributed to increased SOD levels [20,39,40,41]. Thus, our results represent an example of a detrimental consequence of an exogenous SOD (bvSOD3), indicating that bvSOD3 can also play an injurious role in the response to envenoming.

The recognition of stress and damage signals by inflammasomes activates caspase-1, which subsequently induces proinflammatory cytokine secretion and cell death called pyroptosis [31]. Activation of the inflammasome by bee venom induces a caspase-1-dependent inflammatory response [18]. In mouse podocytes, H_2_O_2_ plays a pivotal role in inflammasome formation and activation, which induces caspase-1 activation and IL-1β production [42]. Here, we show that bvSOD3 increased the levels of caspase-1 activity and cell death. Thus, it appears that H_2_O_2_ overproduction by bvSOD3 activates caspase-1, which leads to the induction of proinflammatory cytokine secretion and cell death.

Previous research has shown that ROS triggers an inflammatory response through the expression of many proinflammatory genes [43,44] and that H_2_O_2_ is a signaling molecule in inflammation [22,23,24,25]. Furthermore, bee venom upregulates the level of proinflammatory mediators and cytokines [16,17]. Venom from animals such as scorpions [33], scorpionfish [45], centipedes [34], and snakes [35,46,47,48,49] induces a local inflammatory response in a proinflammatory mediator- and cytokine-dependent manner. Herein, we observed an elevation in the secretion of proinflammatory mediators and cytokine, including TNF-α, COX-2, IL-1β, and IL-6, which was attributed to bvSOD3. Thus, the observed increase in the inflammatory response due to venom administration in the present study was likely caused by bvSOD3. These data indicate that bvSOD3 plays a critical role in the induction of an immediate inflammatory response to venom injection via H_2_O_2_ overproduction. Notably, these findings provide a functional link between bvSOD3 and acute inflammation. In addition, SOD activity in the venom of venomous arthropods suggests a possible SOD-dependent regulation of arthropod venom-induced inflammation.

Although PLA_2_ is regarded as the major bee venom component affecting the inflammatory response [14,15,16], previous studies have shown that melittin is also likely to play a central role in the inflammatory response, including caspase-1 activation and proinflammatory cytokine secretion [16,17]. In addition, similar to previous results [13,14,15,16,17], we found that mice injected with melittin and/or PLA_2_ exhibit an inflammatory response. Importantly, we established that bvSOD3 induced more severe inflammatory effects, as indicated by the increased H_2_O_2_ production, proinflammatory cytokine secretion, and pathological outcome in mice injected with a combination of bvSOD3 and melittin and/or PLA_2_. These findings indicate that bvSOD3 participates in an inflammatory signaling event, in which bvSOD3 generates an inflammation-signaling molecule (H_2_O_2_) to activate caspase-1 and proinflammatory cytokine secretion, promoting an acute inflammatory response.

We describe a crucial role for bvSOD3 in the promotion of the acute inflammatory response and provide a novel view of the mechanism underlying bee-venom-induced acute inflammation. We have incorporated our findings into a proposed model (Figure 3E) to explain the mechanisms underlying the immediate local inflammatory response induced by bvSOD3: bvSOD3, as an exogenous SOD, induces caspase-1 activation and proinflammatory molecule secretion through H_2_O_2_ overproduction using superoxide endogenously produced by melittin and PLA_2_, which promotes an acute inflammatory response. Collectively, bee venom has an ROS-based harm-inducing system for the induction of acute inflammatory responses. Bee venom contains SOD3, which induces H_2_O_2_ and, therefore, can promote acute inflammatory responses. Thus, our data reveal why bvSOD3 exists in the venom and how bee venom acutely elicits inflammatory responses, which suggests that venom-induced acute inflammation is a rapid anti-predatory defense strategy against vertebrate predators.

In contrast, the noxious effects of bvSOD3 in mice elevated type 2 immune responses as a host defense against venoms [17,36]. Our data, together with the observations that catalase and peroxidase activities in the mice injected with AmV increased in a H_2_O_2_-dependent manner and type 2 cytokines were produced in a similar manner to type 1 cytokines, indicate that H_2_O_2_ induced by bvSOD3 in mice is converted by antioxidant system of the host [22] and elevation of the type 2 innate immune responses is a damaged muscle repair mechanism [17,37]. We also detected bvSOD3- and bvPLA_2_-neutralizing IgG antibodies in mice and observed the protection effects in mice immunized with AmV or AmSOD3, demonstrating that immunization with bvSOD3 induces a protective immune response against bee-venom-induced inflammation. These results indicate that bvSOD3 is a toxic component of bee venom and induces an innate immune response and protective immune response, as demonstrated for bee venom PLA_2_ [18].

## 4. Conclusions

Our findings provide the first evidence that bee venom contains the SOD3 enzyme, which promotes the immediate local inflammatory response via H_2_O_2_ overproduction at the site of the sting. To the best of our knowledge, the present study is the first to demonstrate that bvSOD3 acts as an ROS-based harm-inducing system in bee venom to promote an acute inflammatory response. These findings provide a novel view of the mechanism underlying bee venom-induced inflammatory responses. Thus, targeting bvSOD3, which promotes an acute inflammatory response, may offer a new approach to improve the clinical efficacy of therapeutic treatments and VT/VIT. Understanding the crucial role of venom components allows for further insights into the inflammatory and allergic responses induced by arthropod venoms.

## 5. Materials and Methods

### 5.1. Arthropods and Venoms

The bees (*Apis mellifera*, *Apis cerana*, *Bombus terrestris*, and *Bombus ignitus*), centipedes (*Scolopendra subspinipes mutilans*), and hornets (*Vespa mandarinia*) used in the present study were supplied by the Department of Agricultural Biology, National Academy of Agricultural Science, Republic of Korea. Fresh bee venom was collected in test tubes by stimulating the stings of worker bees on the inside wall of the tube. For the immunoprecipitation and injection experiments, *A. mellifera* venom was purchased from Sigma (St. Louis, MO, USA). Spider (*Latrodectus mactans*) venom was also purchased from Sigma. Fresh *S. s. mutilans* venom was collected in test tubes by stimulating the venom forcipules using forceps, and fresh *V. mandarinia* venom was collected by stimulating the sting on the inside wall of the tube.

### 5.2. Cloning and Sequence Analysis

*A. cerana SOD3* (*AcSOD3*) cDNA was selected from a cDNA library, constructed using the entire body of *A. cerana* [50]. Total RNA was extracted from the bees using a Total RNA Extraction Kit (Promega, Madison, WI, USA) and was used as the template for reverse transcription (RT)-PCR. The cDNAs for *A. mellifera SOD3* (*AmSOD3*), *AcSOD3*, *B. ignitus SOD3* (*BiSOD3*), and *B. terrestris SOD3* (*BtSOD3*) were amplified from the total RNA by RT-PCR. The primers for *AmSOD3* cDNA were designed using the following *AcSOD3* cDNA sequences: forward 5′-ATGAATAGAATAATTATA-3′ and reverse 5′-TTTTTTTTTTTTTTTTTT-3′. The primers for the *BtSOD3* and *BiSOD3* cDNAs were designed using the following *BtSOD3* cDNA sequences (GenBank accession number XM_003397268): forward 5′-ATGAATCGAATAATCATA-3′ and reverse 5′-TTTTTTTTTTTTTTTTTT-3′. All PCR products were verified by DNA sequence analysis. The DNASIS and BLAST programs (http://www.ncbi.nlm.nih.gov/BLAST (accessed on 17 July 2016)) were used for pairwise sequence comparisons. MacVector (ver. 6.5, Oxford Molecular, Ltd., Oxford, UK) was used to align the SOD3 amino acid sequences. The signal sequence was predicted using the SignalP 4.1 program (http://www.cbs.dtu.dk/services/SignalP (accessed on 17 July 2016)). The bee SOD3 amino acid sequences registered in this study included AcSOD3 (GenBank accession number KX113616), AmSOD3 (KX113617), BiSOD3 (KX113618), and BtSOD3 (KX113619).

### 5.3. Protein Expression and Analysis

In this study, the recombinant AmSOD3 proteins were produced using a baculovirus expression vector system [51]. The *AmSOD3* cDNA sequence was amplified by PCR from *pAmSOD3* using forward (1–22) and reverse (511–531) primers with the sequences 5′-GGATCCATGAATAGAATAATTATATTAC-3′ and 5′-TCTAGACTAATGATGATGATGATGATGGATAGCTTCGATAATACCACA-3′, respectively, which included a His-tag sequence. After verification of the cDNA sequence, the *AmSOD3* cDNA was inserted into a *pBacPAK8* transfer vector (Clontech, Palo Alto, CA, USA) to generate an expression vector that can drive the expression of recombinant AmSOD3 under the control of the *Autographa californica* nucleopolyhedrovirus (AcNPV) polyhedrin promoter. The recombinant baculoviruses were propagated in Sf9 insect cells cultured in TC100 medium (Gibco BRL, Gaithersburg, MD, USA) at 27 °C [52]. The recombinant proteins were purified using the MagneHis^TM^ Protein Purification System (Promega), and the protein concentrations were estimated using a Bio-Rad Protein Assay Kit (Bio-Rad, Hercules, CA, USA). The recombinant proteins were identified by 12% sodium dodecyl sulfate-polyacrylamide gel electrophoresis (SDS-PAGE). Western blot analyses were performed with an enhanced chemiluminescence western blotting detection system (Amersham Biosciences, Piscataway, NJ, USA) using anti-His antibodies (diluted 1:500 (*v*/*v*); BETHYL Laboratories, TX) or anti-AmSOD3 antibodies [diluted 1:1000 (*v*/*v*)]. Horseradish peroxidase-conjugated anti-mouse IgG (diluted 1:5000 (*v*/*v*); Sigma-Aldrich) was used as a secondary antibody.

### 5.4. Antibody Preparation and Immunoprecipitation

Eight-week-old BALB/c mice (Samtako Bio Korea Co., Osan, Republic of Korea) were injected with a volume of 200 μL, consisting of purified recombinant AmSOD3 (5 μg) mixed with Freund’s complete adjuvant (Sigma-Aldrich). One and two weeks after the first injection, the mice were administered injections of the antigen mixed with Freund’s incomplete adjuvant in a total volume of 200 μL. Blood was collected three days after the final injection (antigen only), allowed to clot overnight at 4 °C, and centrifuged at 10,000× *g* for 10 min. The supernatant antibodies were stored at –70 °C until use. The anti-AmSOD3 antibodies were purified with an A-Sepharose 4B column (Pharmacia) using 0.1 M sodium phosphate buffer (pH 8.0) for binding and 0.1 M sodium citrate buffer (pH 3.5) for elution. For the immunoprecipitation experiments, *A. mellifera* venom (50 μg; Sigma-Aldrich) was incubated with 1, 5, or 10 μg of anti-AmSOD3 antibodies at 37 °C for 1 h. The *A. mellifera* bee venom containing blocked SOD3 (AmV∆SOD3) was determined by SOD activity.

### 5.5. Enzyme Activity Assay

Enzyme activity assays were performed using bee or arthropod venom. Additionally, mouse muscle tissues were ground with liquid nitrogen and centrifuged at 10,000× *g* for 10 min to remove cell debris, and the resulting samples were used for enzyme activity assays. The SOD enzyme activities of bee venom, arthropod venom, and the tissue samples were determined using a Superoxide Dismutase (SOD) Activity Assay Kit (BioVision Inc., Milpitas, CA, USA) according to the manufacturer’s instructions. The amount of SOD3 in bee venom was quantified using a Superoxide Dismutase (SOD) Activity Assay Kit (BioVision Inc.). The catalase enzyme activity of bee venom or the tissue samples was assayed using a Catalase Activity Colorimetric/Fluorometric Assay Kit (BioVision, Inc.) according to the manufacturer’s instructions. The peroxidase enzyme activity was evaluated using a Peroxidase Activity Colorimetric/Fluorometric Assay Kit (BioVision, Inc.) according to the manufacturer’s instructions.

### 5.6. ROS Measurement

To measure the ROS and H_2_O_2_ levels, mouse muscle tissues were prepared as described above. The ROS levels were quantified using an OxiSelect^TM^ In Vitro ROS/RNS Assay Kit (Green Fluorescence; Cell Biolabs, Inc., San Diego, CA, USA) according to the manufacturer’s instructions. The H_2_O_2_ levels were determined using a Hydrogen Peroxide Assay Kit (Abcam, Cambridge, UK) following the manufacturer’s instructions. For the in vitro H_2_O_2_ production of *A. mellifera* bee venom (AmV), recombinant AmSOD3, or AmVΔSOD3, superoxide anions were produced using WST solution (Superoxide Dismutase (SOD) Activity Assay Kit (BioVision Inc.)), which includes xanthine oxidase. The solution was mixed with AmV, recombinant AmSOD3, or AmVΔSOD3. After incubation for 20 min at 37 °C, the H_2_O_2_ levels produced by SOD were determined using a Hydrogen Peroxide Assay Kit (Abcam), as described above.

### 5.7. Venom Administration in Mice

The in vivo experiments were performed using 8-week-old male BALB/c mice purchased from Samtako Bio Korea, Co. (Osan, Republic of Korea). The experimental protocol for the animals was approved by the Dong-A University Animal Care Committee (approval number DIACUC-16-3). The mice were separated into seven groups (5 mice/group): control (PBS), recombinant AmSOD3 (50 or 100 ng per mouse), PLA_2_ (10 μg per mouse), melittin (25 μg per mouse), AmV (50 μg per mouse), or AmVΔSOD3 (50 μg per mouse). In the present study, an inflammatory response was observed in mice after the injection of 50 µg of bee venom, which represents the minimum quantity of venom protein (50–140 µg) released from a sting [53,54]. The amounts of PLA_2_ and melittin in the injection were determined as percentages based on the weight of the dry bee venom [2,6]. PLA_2_ and melittin were isolated from the crude venom fractions by size-exclusion column chromatography, as previously described [55]. The amount of recombinant AmSOD3 in the injection was determined as the quantity (approximately 33 ng) of SOD enzyme in 50 µg of bee venom using the standard curve from the enzyme-linked immunosorbent assay. The mice were shaved, and bee venom or PBS was injected into the inguinal muscle of the rear leg. The venom administration sites were photographed at various timepoints post-injection. The muscle tissue (without skin) at the administration sites was dissected and washed with PBS. The collected muscle samples were used directly.

### 5.8. Apoptosis Assay

The mice’s apoptotic responses to PBS (negative control), AmV (50 μg per mouse), or AmVΔSOD3 (50 μg per mouse) were determined through the measurement of caspase-1 activity using a Caspase-1 Assay Kit (Abcam). As described above, the muscle samples collected at 1, 3, 6, 12, 18, or 24 h post-injection were washed with PBS, resuspended in cold lysis buffer, and placed on ice for 20 min. The lysed samples were centrifuged at 14,000× *g* for 15 min, and the supernatants were mixed with a caspase substrate (YVAD-AFC) in a 96-well plate. The samples were incubated overnight at 37 °C, and the released AFC (7-amino-4-trifluoromethyl coumarin) levels were determined by measuring the absorbance at 400/505 nm using a fluorescence microplate reader (SpectraMAX Gemini XPS, Molecular Devices, Sunnyvale, CA, USA). The experiments were performed in triplicate. The muscle samples collected at 1 and 3 h post-injection were also subjected to immunofluorescence staining. Specifically, the muscle samples were double-labeled with an in situ cell death detection kit (Roche Applied Science) and a mouse anti-caspase-1 antibody (Abcam). The samples were washed three times with PBS and then preincubated in PBS containing 5% BSA at room temperature for 1 h. After incubation for 1 h with mouse anti-caspase-1 antibody (diluted 1:200 (*v*/*v*)) in PBS containing 1% BSA, the samples were washed three times with PBS for 1 h and then incubated with tetramethyl rhodamine isothiocyanate-conjugated goat anti-mouse IgG (diluted 1:300 (*v*/*v*); Santa Cruz Biotech, Inc.) in PBS containing 1% BSA for 3 h. After three washes with PBS, the samples were incubated in a TUNEL reaction mixture containing terminal deoxynucleotidyl transferase and fluorescein-conjugated dUTP at 37 °C for 1 h. The samples were washed with PBS and wet-mounted, and the presence of caspase-1 and apoptosis in the muscle samples was visualized by laser-scanning confocal microscopy (Carl Zeiss LSM 510).

### 5.9. Enzyme-Linked Immunosorbent Assay (ELISA)

The levels of cyclooxygenase (COX)-2, IL-1β, IL-6, tumor necrosis factor (TNF)-α, IL-4, and IL-13 were determined using ELISA Kits (Abcam). The proteins obtained from the mouse muscle samples (50 μg/well) were added to each well of a 96-well plate coated with anti-mouse COX-2, IL-1β, IL-6, TNF-α, IL-4, or IL-13 antibody. After incubation, the wells were washed four times with 300 μL of washing buffer. Subsequently, 100 μL of biotinylated anti-mouse COX-2, IL-1β, IL-6, TNF-α, IL-4, or IL-13 antibody was added to each well, and the plate was incubated for 1 h at room temperature with gentle shaking. The plate was then washed repeatedly, and 100 μL of TMB One-Step Substrate Reagent was added to each well. The plate was subsequently incubated for 30 min at room temperature in the dark with gentle shaking. The reaction was stopped by the addition of 50 μL of stop solution, and the absorbance at 450 nm was immediately measured using a microplate reader (Bio-Rad Model 3550, Bio-Rad, Hercules, CA, USA).

### 5.10. Protection Assay

Eight-week-old male BALB/c mice were immunized through one subcutaneous injection of PBS (control), recombinant AmSOD3 (250 ng per mouse), AmV (50 μg per mouse), or AmVΔSOD3 (50 μg per mouse) and challenged with AmV (50 μg per mouse) one month after the injection. Blood was collected at 1, 3, 6, 12, or 18 h post-challenge, allowed to clot overnight at 4 °C, and centrifuged at 10,000× *g* for 10 min. The supernatant antibodies were used as the primary antibody (diluted 1:500 (*v*/*v*)) for the Western blots. For the Western blot analysis, recombinant AmSOD3 and PLA_2_ proteins were separated by SDS-PAGE on a 12% gel. The Western blot analysis was performed using an enhanced chemiluminescence western blotting detection system with horseradish peroxidase-conjugated goat anti-mouse antibody (diluted 1:5000 (*v*/*v*); Sigma) as the secondary antibody.

### 5.11. Statistical Analysis

Data are shown as the means ± SDs and Shapiro–Wilk test was used to test for normal distribution. The data were analyzed using an independent, unpaired, 2-tailed Student’s *t*-test. All statistical analyses were performed using SPSS PASW 22.0 package for Windows (IBM, Chicago, IL, USA). The statistical significance was set at ** *p* < 0.01 and * *p* < 0.05.

## Figures and Tables

**Figure 1 toxins-14-00558-f001:**
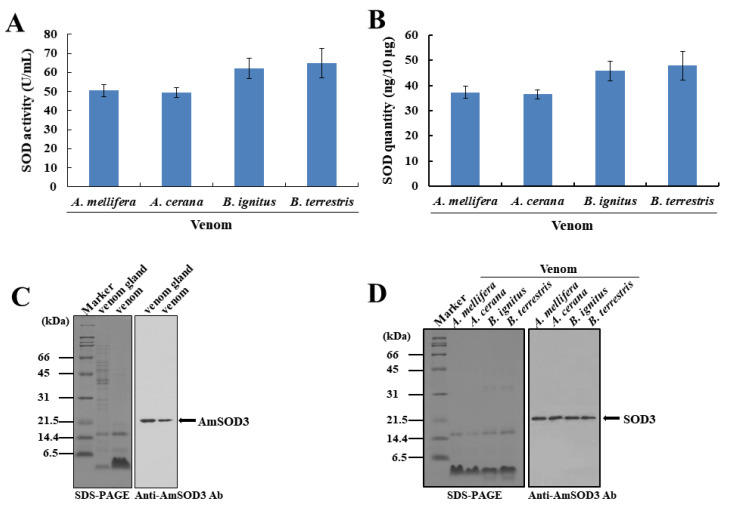
The SOD3 enzyme is a conserved component of venoms from venomous arthropods. (**A**,**B**) SOD activity (**A**) and SOD3 quantities (**B**) in fresh bee venoms. (**C**) Western blot detection of AmSOD3 in the venom glands and venoms of *A. mellifera* worker bees. The results are representative of two independent experiments with different samples. (**D**) Cross-reactivity of the SOD3 enzymes in bee venoms from *A. mellifera*, *A. cerana*, *B. ignitus*, and *B. terrestris*, as analyzed by Western blotting. The results are representative of two independent experiments. (**E**) SOD activity in hornet, spider, and centipede venoms. (**A**,**B**,**E**) The data are shown as the means ± SDs (*n* = 3). The experiments were independently replicated three times.

**Figure 2 toxins-14-00558-f002:**
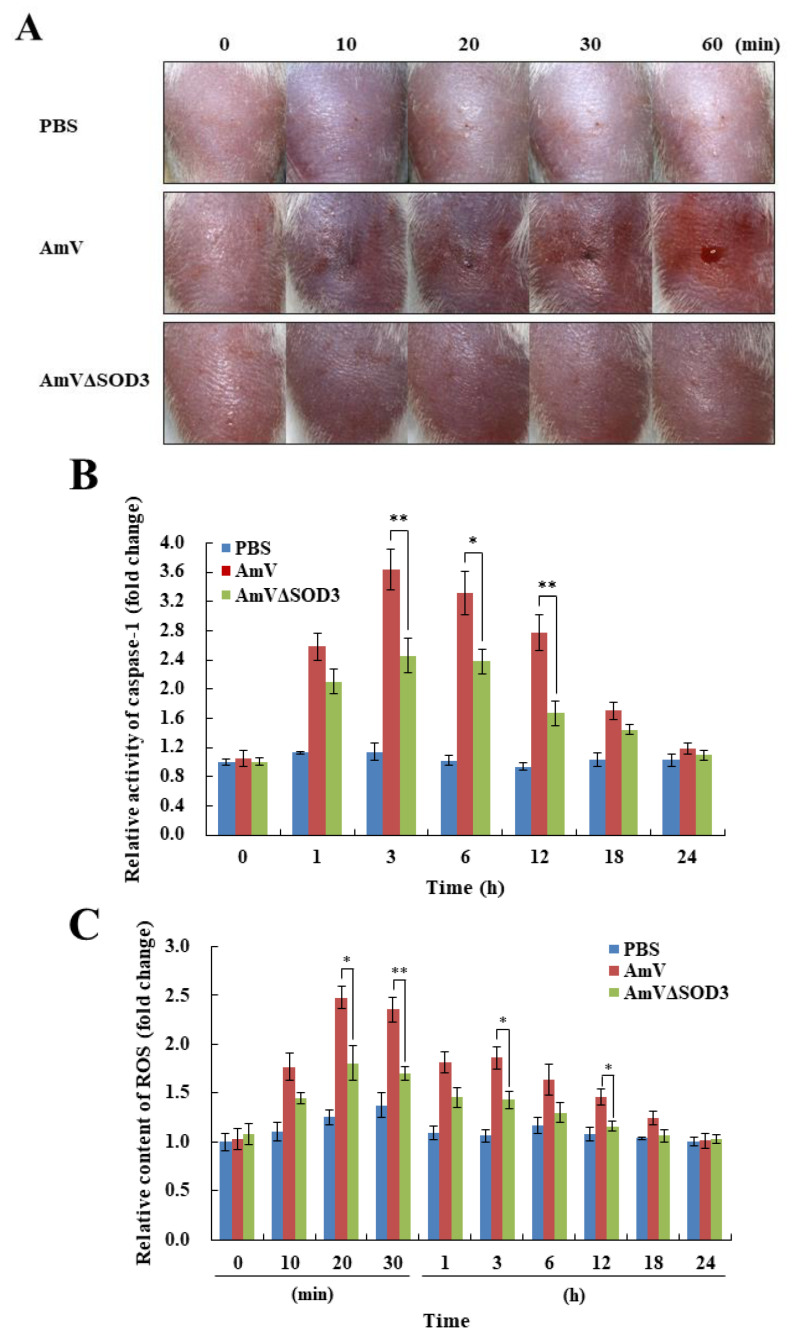
bvSOD3 acts as a ROS-based harm-inducing system to promote acute inflammation. (**A**) Pathological effects of bvSOD3 on the immediate local inflammatory response in mice injected with PBS (injection control), AmV, or AmVΔSOD3, as determined through the imaging of representative mice (group medians) at the indicated timepoints (*n* = 5). (**B**) Caspase-1 activity in the muscles of the mice shown in (**A**). (**C**,**D**) ROS content (**C**) and H_2_O_2_ concentration (**D**) in the muscles of the mice shown in (**A**). (**E**) TNF-α, COX-2, IL-1β, and IL-6 levels produced in the muscles of the mice shown in (**A**), as determined by ELISA. (**B**–**E**) The data are shown as the means ± SDs; the experiments were independently replicated three times. Independent, unpaired, 2-tailed Student’s *t*-test; * *p* < 0.05, ** *p* < 0.01.

**Figure 3 toxins-14-00558-f003:**
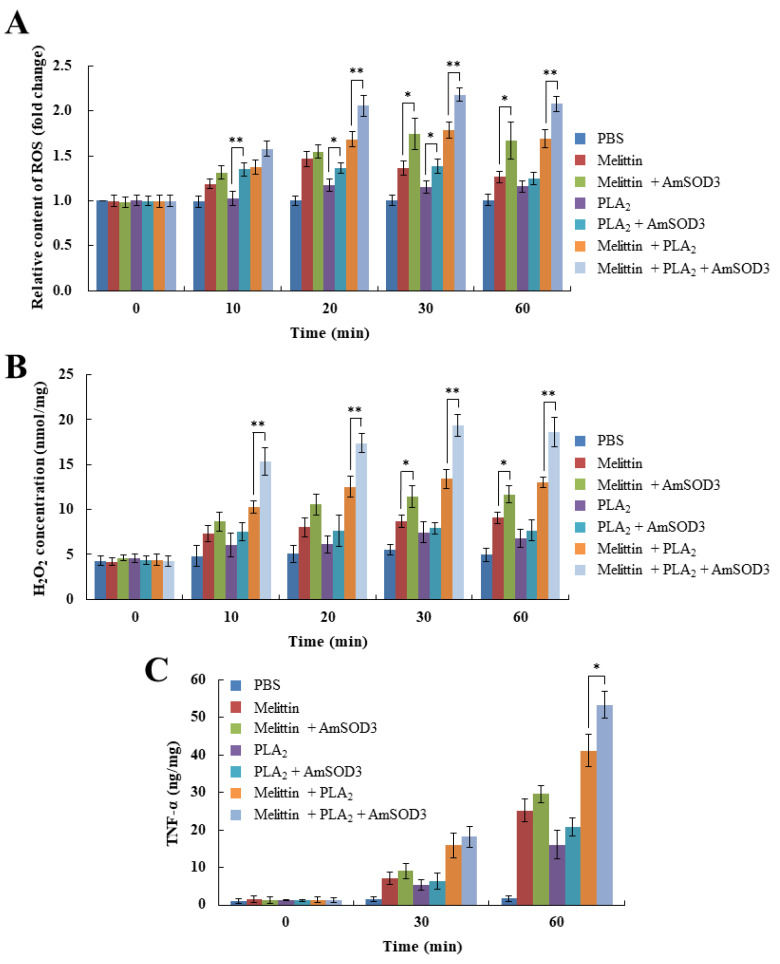
Bee-venom-induced acute inflammation is regulated in a bvSOD3-dependent manner. (**A**,**B**) ROS content (**A**) and H_2_O_2_ concentration (**B**) in the muscles of mice injected with PBS (injection control), major venom components (melittin and/or PLA_2_), or major venom components plus AmSOD3 (*n* = 5). (**C**) Produced levels of TNF-α, COX-2, IL-1β, and IL-6 in the muscles of the mice shown in (**A**), as determined by ELISA. (**D**) Promotion of acute inflammatory responses in the mice shown in (**A**), as determined by the imaging of representative mice (group medians) at the indicated time points (*n* = 5). (**E**) Model of the bee venom-induced acute inflammatory response. (**A**–**C**) The data are shown as the means ± SDs; the experiments were independently replicated three times. Independent, unpaired, 2-tailed Student’s *t*-test; * *p* < 0.05, ** *p* < 0.01.

**Figure 4 toxins-14-00558-f004:**
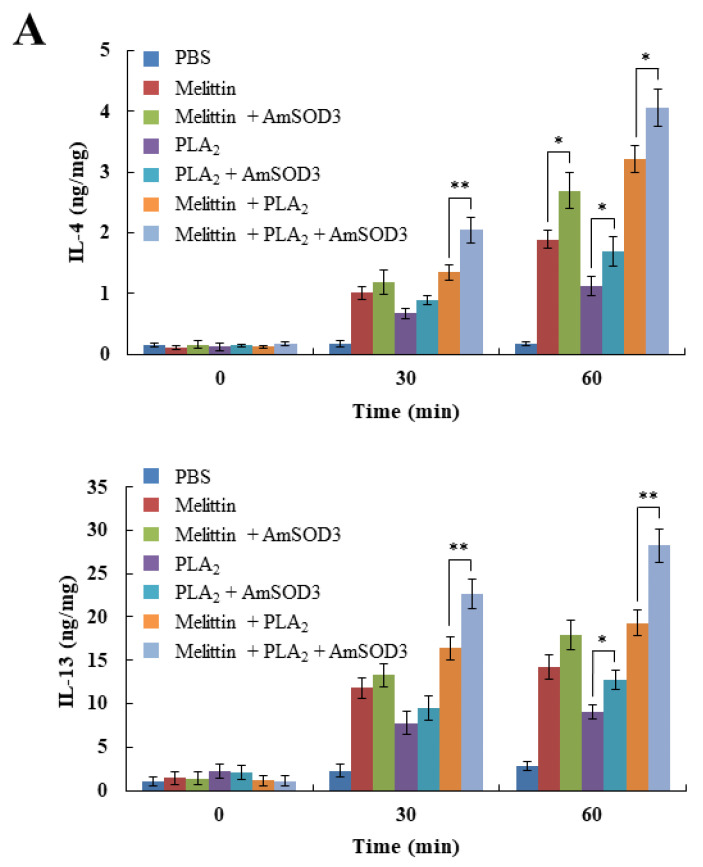
Noxious effects of bvSOD3 elevate the levels of type 2 cytokines. (**A**,**B**) Produced levels of IL-4 and IL-13 in the muscles of mice injected with major venom components (melittin and/or PLA_2_) or major venom components plus AmSOD3 (**A**) or with AmV or AmVΔSOD3 (**B**), as determined by ELISA (*n* = 5). The data are shown as the means ± SDs; the experiments were independently replicated three times. Independent, unpaired, 2-tailed Student’s *t*-test; * *p* < 0.05, ** *p* < 0.01.

**Figure 5 toxins-14-00558-f005:**
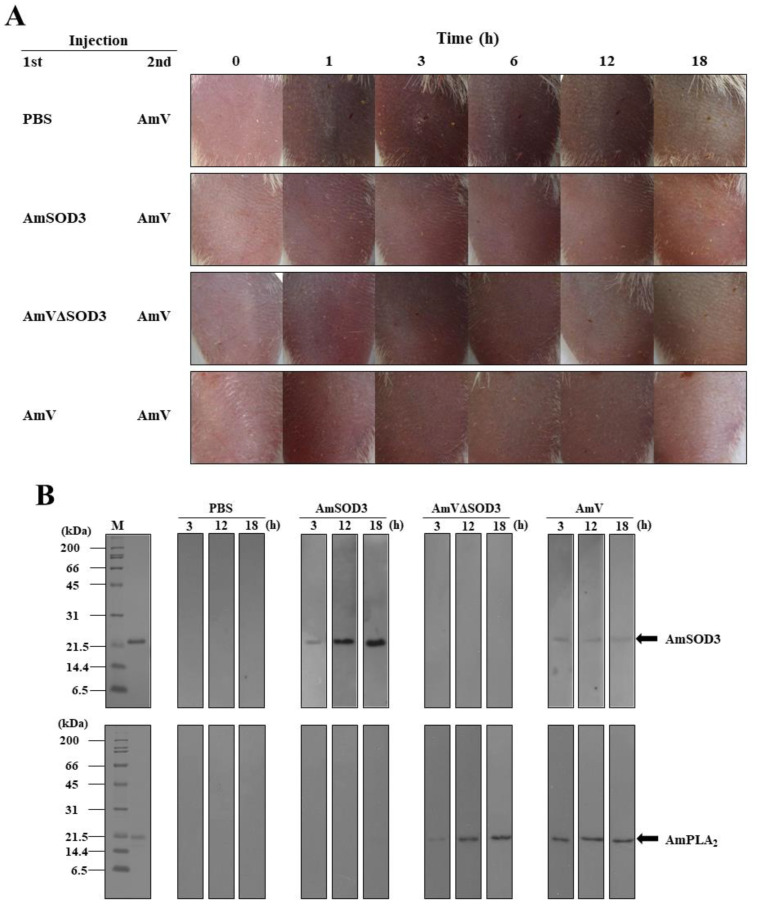
bvSOD3 immunization protects against bvSOD3-mediated toxicity. (**A**) Effects of immunization with bvSOD3 on the acute inflammatory response in mice, as determined through the imaging of representative mice (group medians) at the indicated timepoints. The mice were immunized with PBS (injection control), AmSOD3, AmVΔSOD3, or AmV and challenged with AmV one month later (*n* = 5). (**B**) Western blot detection of anti-AmSOD3 (top) and anti-PLA_2_ (bottom) antibodies in blood samples from the mice shown in (**A**). The results are representative of two independent experiments. (**C**) H_2_O_2_ concentration in the muscles of the mice shown in (**A**). (**D**) Caspase-1 activity in the muscles of the mice shown in (**A**). (**E**) Produced levels of TNF-α, COX-2, IL-1β, and IL-6 in the muscles of the mice shown in (**A**), as determined by ELISA. (**C**–**E**) The data are shown as the means ± SDs; the experiments were independently replicated three times. Independent, unpaired, 2-tailed Student’s *t*-test; * *p* < 0.05, ** *p* < 0.01.

## Data Availability

Not applicable.

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
