# Peer review of "Bee Venom Induces Acute Inflammation through a H2O2-Mediated System That Utilizes Superoxide Dismutase"

_toxins, 2022, doi:10.3390/toxins14080558_

Round 1
Reviewer 1 Report
The article is very interesting. Basically, it brings the novelty of the presence of SOD3 in bee and arthropod-derived venoms. The results are highly innovative, indicating the toxic and inflammatory role of this component. Congratulations to the authors! However, some points should be adjusted and even better explained/discussed before the article can be published.
Introduction
1. “Incidences” is with diferente letter
2. Change envenomations for envenomings as recommended by WHO.
3. I recommend author read the paper https://www.frontiersin.org/articles/10.3389/fimmu.2019.02090/full . Informations are wrong in the manuscript, such as:
….The predominant allergen of bee venom is melittin, which possesses lytic activity, followed by phospholipase A2 (PLA2), which has inflammatory and nociceptive effects…
The predominant allergen is PLA2, mellitin in the major toxic compound.
4. In addition, regarding this sentence: ….nor the component that promotes bee venom-induced immediate inflammation is known…..I also recommend author to check the article - https://pubmed.ncbi.nlm.nih.gov/32457615/
5. I do not agree. Please rewrite the sentence below.
Because bee venom serves as a defensive weapon against human and vertebrate predators, we hypothesized that SOD serves as an ROS-based harm-inducing system in bee venom, rather than a defensive system against ROS.
6. Check English in all the Manuscript. There are few weird phrases.
Results
1. Change "contained" by "present"
2. How authors can confirm that they block (immunoprecipitation) all the SOD3?
3. Do not use “almost no H2O2”- - maybe reduced levels…..
4. Figure 2A is only demonstrative, since only one animal per group was used. There are many other ways to measure redness and edema. Please make it clear. Same for Figure 3D and 5..
5. See the sentence below:
Collectively, these results indicate that bvSOD3 promotes acute induction of inflammatory responses driven by melittin and PLA2 [12,14] and thus provide a rationale for the presence of SOD3 in bee venom.
Authors should study and discuss the synergism mechanism of venom compounds - https://pubmed.ncbi.nlm.nih.gov/32457615/
Figure 3 reflects synergism mechanism.
6. In respect to Th1 and Th2 response. It is clear that both types are activated by bee venom since authors identified levels of both in mice. However, we cannot conclude that it was based only in bvSOD3. Additional experiments are necessary. We know that one T cell response can equilibrate the other. Just make sure to not overestimate results.
After improve results, please review and realign the discussion and abstract.
Author Response
#1 Comments and Suggestions for Authors
The article is very interesting. Basically, it brings the novelty of the presence of SOD3 in bee and arthropod-derived venoms. The results are highly innovative, indicating the toxic and inflammatory role of this component. Congratulations to the authors! However, some points should be adjusted and even better explained/discussed before the article can be published.
Introduction
- “Incidences” is with diferente letter
- Change envenomations for envenomings as recommended by WHO.
Author’s response
Thank you very much for kind comment on that. According to reviewer #1’s comment, we corrected it.
- I recommend author read the paper https://www.frontiersin.org/articles/10.3389/fimmu.2019.02090/full . Informations are wrong in the manuscript, such as:
….The predominant allergen of bee venom is melittin, which possesses lytic activity, followed by phospholipase A2 (PLA2), which has inflammatory and nociceptive effects…
The predominant allergen is PLA2, mellitin in the major toxic compound.
Author’s response
We are very grateful to you for pointing our mistakes. According to Reviewer’s comments, we revised the sentence in the Introduction section and cited the reference suggested by reviewer.
- In addition, regarding this sentence: ….nor the component that promotes bee venom-induced immediate inflammation is known…..I also recommend author to check the article - https://pubmed.ncbi.nlm.nih.gov/32457615/
Author’s response
We are very grateful to you for valuable comments. According to Reviewer’s comments, we revised the part in the Introduction section and cited the reference suggested by reviewer.
- I do not agree. Please rewrite the sentence below.
Because bee venom serves as a defensive weapon against human and vertebrate predators, we hypothesized that SOD serves as an ROS-based harm-inducing system in bee venom, rather than a defensive system against ROS.
Author’s response
We are very grateful to you for valuable comments. According to Reviewer’s comments, we revised the sentence in the Introduction section.
- Check English in all the Manuscript. There are few weird phrases.
Author’s response
Thank you very much for kind comment on that.
Results
- Change "contained" by "present"
Author’s response
Thank you very much for kind comment on that. According to reviewer #1’s comment, we corrected it.
- How authors can confirm that they block (immunoprecipitation) all the SOD3?
- Do not use “almost no H2O2”- - maybe reduced levels…..
Author’s response
We are very grateful to you for valuable comments. As shown in Fig S2A and S2B, the immunoprecipitation results revealed that AmΔSOD3 produced almost no H2O2.
- Figure 2A is only demonstrative, since only one animal per group was used. There are many other ways to measure redness and edema. Please make it clear. Same for Figure 3D and 5..
Author’s response
We are very grateful to you for valuable comments. As described in Figure legends, the results were determined through the imaging of representative mice (group medians).
- See the sentence below:
Collectively, these results indicate that bvSOD3 promotes acute induction of inflammatory responses driven by melittin and PLA2 [12,14] and thus provide a rationale for the presence of SOD3 in bee venom.
Authors should study and discuss the synergism mechanism of venom compounds - https://pubmed.ncbi.nlm.nih.gov/32457615/
Figure 3 reflects synergism mechanism.
Author’s response
We are very grateful to you for valuable comments. According to Reviewer’s comments, we revised the sentence in the Results section and cited the reference suggested by reviewer.
- In respect to Th1 and Th2 response. It is clear that both types are activated by bee venom since authors identified levels of both in mice. However, we cannot conclude that it was based only in bvSOD3. Additional experiments are necessary. We know that one T cell response can equilibrate the other. Just make sure to not overestimate results.
Author’s response
We are so sorry about that and thank you very much for kind comment on that. Thus, further investigation will consider your valuable suggestion.
After improve results, please review and realign the discussion and abstract.
Author’s response
We are very grateful to you for valuable comments.
Reviewer 2 Report
It is an interesting study, very detailed in materials and methods and in description of possible mechanisms of inflammatory cascade induced by superoxide dismutase (SOD3). Certainly, it provide a novel view of the mechanism underlying bee-venoma-induced acute inflammation but further studies are necessary to establish if this findings could be useful to offer a new approach for bee venom preparations for immunotherapy.
In my opinion, this is a preliminar study with uncertain future clinical applications; although this consideration, because of its high quality this work could be approved without any changes.
Author Response
#2 Comments and Suggestions for Authors
It is an interesting study, very detailed in materials and methods and in description of possible mechanisms of inflammatory cascade induced by superoxide dismutase (SOD3). Certainly, it provide a novel view of the mechanism underlying bee-venoma-induced acute inflammation but further studies are necessary to establish if this findings could be useful to offer a new approach for bee venom preparations for immunotherapy.
In my opinion, this is a preliminar study with uncertain future clinical applications; although this consideration, because of its high quality this work could be approved without any changes.
Author’s response
We are very grateful to you for valuable comments.